# Experimental Investigation into the Effect of Pyrolysis on Chemical Forms of Heavy Metals in Sewage Sludge Biochar (SSB), with Brief Ecological Risk Assessment

**DOI:** 10.3390/ma14020447

**Published:** 2021-01-18

**Authors:** Binbin Li, Songxiong Ding, Haihong Fan, Yu Ren

**Affiliations:** 1College of Materials Science and Engineering, Xi’an University of Architecture and Technology, Xi’an 710055, China; libinbin1987@xauat.edu.cn (B.L.); RENYU612323@163.com (Y.R.); 2Department of Civil Engineering, Faculty of Science and Engineering, University of Agder, N4879 Grimstad, Norway

**Keywords:** sewage sludge, pyrolysis, ecological risk assessment, BCR sequential extraction, heavy metals

## Abstract

Experimental investigations were carried out to study the effect of pyrolysis temperature on the characteristics, structure and total heavy metal contents of sewage sludge biochar (SSB). The changes in chemical forms of the heavy metals (Zn, Cu, Cr, Ni, Pb and Cd) caused by pyrolysis were analyzed, and the potential ecological risk of heavy metals in biochar (SSB) was evaluated. The conversion of sewage sludge into biochar by pyrolysis reduced the H/C and O/C ratios considerably, resulting in stronger carbonization and a higher degree of aromatic condensation in biochar. Measurement results showed that the pH and specific surface area of biochar increased as the pyrolysis temperature increased. It was found that elements Zn, Cu, Cr and Ni were enriched and confined in biochar SSB with increasing pyrolysis temperature from 300–700 °C; however, the residual rates of Pb and Cd in biochar SSB decreased significantly when the temperature was increased from 600 °C to 700 °C. Measurement with the BCR sequential extraction method revealed that the pyrolysis of sewage sludge at a suitable temperature transferred its bioavailable/degradable heavy metals into a more stable oxidizable/residual form in biochar SSB. Toxicity of heavy metals in biochar SSB could be reduced about four times if sewage sludge was pyrolyzed at a proper temperature; heavy metals confined in sludge SSB pyrolyzed at about 600 °C could be assessed as being low in ecological toxicity.

## 1. Introduction

The amount of urban sewage sludge has increased dramatically in recent years due to the rapid urbanization. The latest statistic data show that the quantity of sewage sludge (water content 80%) in China reached 90 million tons in 2020 [1,2]. Without being properly treated, various toxic and harmful substances in sewage sludge, such as parasite eggs, pathogenic microorganisms and heavy metals, could cause pollution of soil and water bodies. Conventional sludge treatment technologies mainly include sanitary landfill and incineration technologies [3,4]. With shortages of land resources, however, sanitary landfill technology has been greatly restricted or even banned in China; for example, urban sludge stabilized by lime can only be used on non-food cropland due to the complexity and variability of the pollutant composition in the sludge. The high cost and the risk of secondary pollution have limited the development of sludge incineration and treatment technology [5].

Cement kiln cooperative sludge disposal technology has been used to pyrolyze sewage sludge [6]. The exit flue gas in cement plants is about 300 °C, while the residual flue gas at the click cooler is about 600–700 °C. This technology is used in cement plants to dry and pyrolyze the sludge. The sludge thus treated can be used to prepare cement clinker [7]. With this technology, there is no need to build a new sludge treatment production line, only a sludge treatment unit, which therefore greatly reduces the investment and operating costs [8,9].

Treatment of heavy metals in sludge by pyrolysis (cement kiln cooperative sludge disposal technology) has increasingly attracted more attention [10,11,12]. Liu [13] studied the migration pattern of heavy metals during hydrothermal carbonization and concluded that the heavy metals in sludge after pyrolysis were mainly concentrated in sludge biochar, which had higher thermal stability than bio-oil. Jin [5] conducted an in-depth study on the chemical behavior and bioavailability of heavy metals after the sludge had been through the process of pyrolysis at 400–600 °C. The study showed that different chemical forms of heavy metals directly affect the toxicity, migration and circulation in nature. Developing a method to assess the environmental risk caused by the migration of heavy metals in biochar is essential for the green utilization of urban sludge.

There is a lack of research on the potential environmental risks of sludge pyrolysis products at 300–700 °C. It is therefore worthwhile and of practical importance to study the effect of pyrolysis temperature on the solidification/stabilization of heavy metals in regard to streaming residual flue gas from, for example, a cement production line, to establish proper heating conditions in pyrolysis. It is also important to examine the existing chemical forms of heavy metals in sludge pyrolysis products in order to carry out their ecological environment risk assessments. The main purposes in the present study were (1) to investigate the variation of physicochemical, structural and morphological properties of biochar at different temperatures; and (2) to evaluate the concentration, chemical forms and residual rates of heavy metals as stabilized and confined in sludge biochar at various pyrolysis temperatures. The potential risk of those stabilized heavy metals in sludge biochar to the environment was also addressed.

## 2. Materials and Methods

### 2.1. Pyrolysis of the Sewage Sludge Biochar (SSB)

A sewage sludge sample was taken from the dehydration workshop of Xi’an Biyuan Water Co., Ltd., China. The sludge sample was dried at 105 °C for 24 h and ground with a sample preparation machine. The particle size of the ground sludge was controlled with a 100-mesh sieve; various samples were prepared and preserved in sealed plastic bags.

The pyrolysis of sludge was completed using the device shown in Figure 1. The main components of this device included a high-pressure nitrogen cylinder, an electrically heated horizontal quartz tube furnace, particle filtration equipment and acetone solution for collecting pyrolysis oil.

In each experiment, 20 g of dry sludge sample was first placed in the square corundum crucible of the furnace, and then air was expelled by blowing nitrogen gas into the crucible at a rate of 1 L/min for 10 min. The quartz tube furnace was then heated to a designated temperature at a rate of 10 °C/min from ambient temperature; the sludge sample were then maintained at this designated temperature in the furnace for 60 min to be converted into biochar SSB (weighted in quality W). Various designated temperatures were selected for research purpose in the present study; they were set at 300 °C, 400 °C, 500 °C, 600 °C and 700 °C.

### 2.2. Measurement of Biochar SSB

The elemental contents of C, H, N and S in sewage sludge and biochar were determined using an organic element analyzer (Vario EL III, Hanau, Germany). The samples of sludge and biochar SSB were diluted with water in a mass ratio of 1:20; its pH value was measured with a pH meter.

The weight ratio of biochar SSB (W) to dry sewage sludge was used to calculate the biochar yield Y (Y = W/20). The ash content in the sample was determined based on the coal industry analysis method (GB/T212-2001). A specific surface area analyzer (ASAP2020, Micromeritics, Norcross, GA, USA) was used to measure the specific surface area of the sample.

Fourier transform infrared spectroscopy (iN10-IZ10, Thermo Fisher, Waltham, MA, USA) was used to determine the functional groups of the sludge and biochar. Field emission electron microscopy (JSM-7610F, JEOL, Tokyo, Japan) was used to analyze the apparent morphology of the samples.

### 2.3. Extraction and Analysis of Heavy Metals

#### 2.3.1. Concentrations and Chemical Forms of Heavy Metals

It is known that heavy metals (Zn, Cu, Cr, Ni, Pb and Cd) in sewage sludge exist in variety of chemical forms, for example, in exchangeable/carbonate bound form, in the oxidizable form or in residual form etc.; pyrolysis would cause those heavy metals undergoing a chemical form change to be confined in biochar SSB as residues, or to some of them, for example Pd and Cr, vaporizing as loss on ignition if the pyrolysis temperature is sufficient high. The heavy metals contained in sewage sludge and those confined in biochar SSB could be sequentially extracted by the three-step sequential extraction method (as specified and modified by the Community Bureau of Reference (BCR), Commission of the European Communities) [14,15,16]. The steps were as follows: (1) for metals of the exchangeable/carbonate bound form (noted as F1), 0.5 g of sludge/SSB sample was loaded in a 50 mL polyethylene centrifuge tube and therein mixed with 20 mL of glacial acetic acid (HAc) solution (0.1 mol/ L). The mixture was centrifuged at 4000 r/min for 20 min after being stirring at room temperature for 16 h. The supernatant was collected and stored in a 50 mL volumetric flask after filtering with 0.45 μm cellulose acetate fiber membrane to be used later to measure the content/score (C_i_) of a specific metal’s potential contamination risk. (2) For metals in a reduced form (F2), 20 mL of hydroxylamine hydrochloride (NH_2_OH·HC1) solution (0.1 mol/L) was added to the solid residue from the last step (i.e., Step 1), and the mixture was stirred at room temperature for 16 h and centrifuged at 4000 r/min for 20 min. The supernatant was collected in the same method as in Step 1 for content/score (C_i_) measurement. (3) For metals in the oxidizable form in sample (F3), a water bath was used; 5 mL of 30% hydrogen peroxide (H_2_O_2_) was added to the solid phase residue from Step 2. This mixture was placed in a water bath at 25 °C with intermittent stirring for 1 h; it was again, after adding another 5 mL of 30% hydrogen peroxide (H_2_O_2_), placed in a water bath at 85 °C with intermittent stirring for another 1 h and then evaporated to be nearly dry in a water bath. Subsequently, this after-evaporated sample was diluted by adding 25 mL of ammonium acetate (NH_4_Ac) solution (1 mol/L; pH was adjusted to 2 with nitric acid) and stirred for 16 h at ambient temperature. It was then centrifuged at 4000 r/min for 20 min. The supernatant was collected in the same method as in Step 1. (4) For the residual form (F4) of heavy metals in the sample, digestion was carried out according to the US EPA 3050B method; the solution to be measured was thus prepared. The heavy metal contents in such solutions were determined using an inductively coupled plasma-optical emission spectrometer (ICP-OES, 715-ES, Varian, Palo Alto, CA, USA).

As the sludge is pyrolyzed, most of the heavy metal is enriched and confined as a residue in biochar SSB [17]. In the present paper, the residual rate R of heavy metals was proposed as a parameter to characterize the curing ratio of heavy metals in the process of sludge biochar generation, and the formula is as follows:(1)$$R\%=\frac{{\mathrm{SSB}}_{X}\times Y}{{\mathrm{SS}}_{X}}\times 100$$
where R is the residual rate of heavy metals (%), subscript x represent a specific heavy metal, SSB_X_ is the total concentration of this heavy metal in biochar SSB (mg·kg^−1^), SS_X_ is the concentration of the same heavy metals in SS (mg·kg^−1^), and Y is the yield of biochar (%).

#### 2.3.2. Ecological Risk Index (RI) of Heavy Metals

Based on the total concentration, quantity, toxicity and sensitivity of heavy metals, Hakanson [18] evaluated the potential risk of heavy metal contamination in the samples. Among them, the scores (C_i_) of the exchangeable/carbonate bound form (F1) and reduced form (F2) represented direct toxicity, the score of oxidizable form (F3) represented potential toxicity, and the score of residual form (F4) represented non-toxicity. The RI could be calculated as follows:(2)$$C_{f}=C_{i}/C_{n}$$
(3)$$E_{r}=T_{r}\cdot C_{f}$$
(4)$$\mathrm{RI}={\Sigma E}_{r}$$
where C_f_ is the pollution factor of a specific heavy metal; C_i_ represents the scores of this metal as measured in the F1 + F2 + F3 section; C_n_ represents the sum of scores of the same metal in F4; T_r_ represents the toxicity reaction factor of a specific heavy metal; available in the literature (the values of each heavy metal were Zn (1), Cu(5), Cr (2), Ni (6), Pb (5) and Cd (30)); E_r_ is the heavy metal pollution and ecological risk factor, obtained by multiplying T_r_ and C_f_ of heavy metal; and the sum of E_r_ is regarded as the potential for overall pollution ecological risk index RI.

The SPSS statistical package was used for statistical analysis of the data. The least significant difference (LSD) method was used to analyze the deviation level of each average value, and the significant level was *p* < 0.05.

## 3. Results and Discussion

### 3.1. The Characteristics of Sewage Sludge and SSB

The parameters to characterize sewage sludge and biochar as produced at different temperatures in a quartz tube furnace are yield (Y), elemental composition, pH value, ash content and specific surface area; they are given in Table 1. As seen, the biochar yield decreased from 74.3% to 51.4%, whereas mass ratio of ash in sewage sludge increased from 38.8% to 82.5% when the pyrolysis temperature increased from 300 °C to 700 °C; this is because the organic components in the sludge were gradually decomposed [13,19]. One may also notice that a loss in sample weight occurred relatively quicker at the temperature ranges of 300–400 °C and 600–700 °C. This indicated that organic matter decomposed rapidly in these temperature ranges, while the inorganic components were concentrated and retained in the biochar. The pH value of sewage sludge was close to neutral, but the pH of the biochar SSB obtained by pyrolysis was alkaline [20]. As the pyrolysis temperature increased, the alkalinity gradually increased, consistent with results reported in the literature. The pH of biochar SSB falls in the alkaline range and is mainly affected by the ash composition on the surface of biochar SSB and the high-aromaticity structure in biochar [17].

With increasing pyrolysis temperature, the elements C, H, N, O and S contents in biochar SSB gradually decreased. Measurements found that the contents of C and H decreased by 17.85% and 31.40%, respectively, when sewage sludge was heated in the temperature range of 300–400 °C, indicating that C and H were mainly converted into volatile substances in the form of H_2_O or hydrocarbon compounds during the pyrolysis process. In the temperature range between 600 and 700 °C, carbon oxides may be formed, leading to decreases of C and O elements. The molar H/C and O/C ratios could be used as carbonization parameters to characterize the organic aromaticity of biochar; those two ratios both decreased considerably with increasing pyrolysis temperature, indicating that the stability of biochar SSB increased with the degree of aromatic condensation, which is consistent with the results reported by Huang et al. [19]. Table 1 also shows that the specific surface area of biochar SSB increased considerably with increases in pyrolysis temperature from 300 to 700 °C compared to that of the original sewage sludge.

### 3.2. FTIR Analysis

The FTIR spectra of sewage sludge and biochar SSB produced at different pyrolysis temperatures are shown in Figure 2. As seen, the absorption peak/dip found in the region of wavenumber 3300–3500 cm^−1^ represented the stretching vibration of radical –OH related to water, alcohol and carboxylic acid. The –OH vibration peak gradually weakened as the pyrolysis temperature increased; this is because an increase in the pyrolysis temperature led to a complete dehydration reaction, and therefore the hydroxyl functional group was gradually broken. The absorption peaks in the region of wavenumber 2600–3000 cm^−1^ were both symmetric and antisymmetric stretching vibrations of CH_x_, indicating that an aliphatic structure existed in sewage sludge. As the temperature increased, this peak intensity decreased and eventually disappeared when the pyrolysis temperature was higher than 500 °C, that is, most of the aliphatic hydrocarbons decomposed into carbon dioxide, methane and other gases or were converted into aromatic structures [21]. The cleavages of –OH and CH_x_ groups observed with the FTIR spectrum between 300–400 °C were consistent with the reductions of H and O.

The absorption peak near the wave number 1650 cm^−1^ represented the stretching vibration of the amide bond (–CONH–). As the pyrolysis temperature increased, its peak intensity decreased slightly. This was due to the decomposition of the amide functional group in the biochar SSB or the complex reaction of the functional group with heavy metals. The absorption peak near the wavenumber 1620 cm^−1^ was the stretching vibration of the aromatic ring C=C. As the pyrolysis temperature increased, the vibration gradually weakened; that is, the sludge was fully decomposed, and the degree of aromatization of the biochar gradually increased, consistent with the H/C ratio changes in Table 1. The strong absorption peak near the wavenumber 1050 cm^−1^ represented the stretching vibration of –CO on the aromatic ring [22]. This peak had slight changes with increasing temperature, which was mainly because the different forms of oxygen in the sludge were converted into carbon chains and carbon–oxygen single bond form; one may conclude that pyrolysis temperature had a minor effect on this process.

### 3.3. SEM Analysis

Figure 3 shows a set of SEM images with 10,000× and 20,000× magnification of sewage and biochar SSB as obtained at different pyrolysis temperatures. As seen from Figure 3a,b, the surface of sewage sludge was smooth and had a non-porous structure; it was mainly because the organic matter formed a smooth, dense structure on the surface. Inorganic particles of approximately 1 μm were attached to the surface. The biochar obtained by pyrolysis at 300 °C (SSB-300, Figure 3c,d) exhibited a terrace-like multi-layer structure with small particles of inorganic compounds distributed on the structure. This was because the fatty organic matter in the surface layer was decomposed into methane, carbon dioxide and other gases that evaporated during the pyrolysis process, causing the volume of the surface layer to be reduced, whereas the decomposition of internal organic components had not yet started as the heat still to be transferred to that region; as a result, a terrace-like structure was formed. In a temperature range between 300 and 500 °C, the surface layer of biochar continued to decrease in size, and the overall size of the organic particulates also decreased (Figure 3e–h). In the biochar as pyrolyzed at 600 °C (SSB-600, Figure 3i,j), it was found that porous structures existed on the surface, while the volume of the entire organic particulate matter was reduced to approximately 5 μm. It confirmed that the main cause affecting the morphology of biochar SSB could be contributed to the shrinkage of the surface layer and formation of tiny pores as the internal volatile gas was released. When the pyrolysis temperature was increased to 700 °C, the biochar structure gradually became small particle clusters, with sizes of about 1 μm (Figure 3k,l); this is mainly because the organic matter had been decomposed at this temperature; as a result, the inorganic matter was no longer bound to the original organic matter.

### 3.4. Analysis of Heavy Metals

#### 3.4.1. Total Contents of Heavy Metals in Sewage Sludge and Biochar SSB

Table 2 shows the contents of the heavy metals Cd, Cr, Cu, Ni, Pb and Zn in sewage sludge and biochar SSB. The content of heavy metal Zn in sewage sludge was 1341.73 mg·kg^−1^. It was considerably higher than that of other heavy metals. In general, the total heavy metal concentrations in sewage sludge and biochar were in a descending order of Zn > Cu > Cr > Pb > Ni > Cd. With increases in pyrolysis temperature, loss of organic mass resulted in an enrichment of heavy metals in biochar. The concentrations of various heavy metals in biochar SSB were higher than those in sewage sludge.

Figure 4 shows the residual rate of heavy metals in biochar at different pyrolysis temperatures. The residual rates of Zn, Cu, Cr and Ni in biochar were 90.21%, 95.02%, 92.97% and 97.57%—most of those heavy metals accumulated and were confined in the biochar SSB after pyrolysis. At the same time, it was found that when the pyrolysis temperature rose to 700 °C, the residual rate of heavy metal Cd in biochar SSB decreased from 90.72% to 79.27% and that of Pb from 91.26% to 86.48% in biochar. It is known that Cd and Pb mainly exist in the form of carbonates in sludge, and that an increase in temperature up to 700 °C causes decomposition of carbonates, resulting in the volatilization of heavy metals. A pyrolysis temperature higher than 700 °C leads to the volatilization of the heavy metals Cd and Pb.

#### 3.4.2. Analysis of the Chemical Forms of Heavy Metals in Sewage Sludge and Biochar SSB

It is well accepted that the heavy metals existing in the exchangeable and acid-soluble form (F1) and reduced form (F2) as extracted by the BCR can be directly degraded by organisms. These are bioavailable heavy metals and are easily leached out. The heavy metals in the oxidizable form (F3) are potentially bioavailable heavy metals. These types of heavy metals can only be degraded and leached out by strong acids or oxidizing environments. The heavy metals in the residual form (F4) are non-bioavailable heavy metals; heavy metals of this type cannot be leached out or degraded.

Figure 5 shows the percentage of the six heavy metals (Cd, Cr, Cu, Ni, Pb and Zn) in sewage sludge and biochar SSB as obtained by the BCR method; as seen, they existed in four chemical forms. The ratios of heavy metals Zn, Cd, Pb and Ni existed in exchangeable and acid-soluble (F1), reducible and reduced forms (F2 + F3) and reached 91.62%, 76.85%, 70.95% and 78.05% in sewage sludge, indicating that a direct discharge of sludge into the environment has a high potential ecological risk. With increases of pyrolysis temperature, it was found that the residual form (F4) content of the heavy metals Zn, Cd, Pb and Ni in biochar increased significantly. When the pyrolysis temperature was 700 °C, contents of Zn, Cd, Pb and Ni in the residual form (F4) could reach 66.81%, 60.34%, 59.53% and 63.79%; Zn, Cd, Pb and Ni were significantly stabilized in biochar SSB, confirming that the conversion of sewage sludge into biochar by pyrolysis at a suitable temperature would transfer some the heavy metals into to more stable residual forms.

Measurement also revealed that the heavy metals Cu and Cr mainly existed in the reduced form (F3) and residual form (F4) in sewage sludge. After sewage sludge was converted into biochar SSB by pyrolysis, the exchangeable and acid-soluble form (F1) and reduced form (F2) of Cu and Cr decreased quickly (even disappeared).

In general, all six heavy metals changed into a more stable form when sewage sludge was converted into biochar by pyrolysis with increasing temperature; identifying a proper pyrolysis temperature is crucial to converting them into a residual form in biochar SSB.

#### 3.4.3. Ecological Risk Assessment of Heavy Metals

The ecotoxicity and bioavailability of heavy metals depends on the amount of heavy metals and, more importantly, on the existing chemical forms of those metals. Based on the calculation given in Section 2.3.2, the pollution factor C_f_, the toxicity reaction factor E_r_ and the ecological risk index RI values were calculated according to the ratio of different chemical forms of each heavy metal. The results are given in Figure 6 and were used to evaluate the environmental risk level of heavy metals in sewage sludge and biochar SSB.

As seen in Figure 6, as an example, the C_f_ value of heavy metal Zn in sewage sludge was 10.93; after pyrolysis at 700 °C, the C_f_ value dropped to 0.5, and the pollution factor caused by Zn changed from a high ecological risk to no ecological risk. In general, the C_f_ value of each heavy metal decreased as the pyrolysis temperature increased. One may assert that the pyrolysis process reduces the ecological risk of heavy metals.

It is obvious in Figure 6c that the RI value of the six heavy metals had a maximum of 155.03 in sewage sludge, which would exert a high environmental risk to soil and water bodies if it was directly discharged to agricultural land. However, the RI value of biochar SSB decreased dramatically with increases in the pyrolysis temperature. It is worth noting that the RI value of biochar after pyrolysis at temperatures greater than 600 °C was less than 50.0, about four-times less than that of sewage sludge. These results illustrated that the environmental risks of heavy metals after pyrolysis at proper temperatures could be expected to be rather limited. Pyrolysis at a high temperature of 600 °C is recommended.

## 4. Conclusions

Sewage sludge was pyrolyzed into biochar SSB at various designated temperatures with a home-made pyrolysis carbonization device; the effect of pyrolysis temperature on the characteristics, structure and total heavy metals (Cd, Cr, Cu, Ni, Pb and Zn) in the pyrolysis product, biochar (SSB), was addressed. It was found that an increase in pyrolysis temperature from 300 to 700 °C caused the aromatization and the increase in specific surface area of biochar, leading to heavy metals being confined in biochar SSB; however, the residual rates of Pb and Cd in biochar decreased significantly when the temperature was increased from 600 °C to 700 °C; more importantly, the pyrolysis of sewage sludge at a suitable temperature (for example at 600 °C) transferred its bioavailable/degradable heavy metals into a more stable oxidizable/residual form in biochar SSB, and a significant reduction in the bioavailability of heavy metals in biochar SSB could be expected. The ecological risk assessment showed that the heavy metals in sewage sludge had a considerably high ecological toxicity factor, which could be reduced by about four times if sewage sludge was pyrolyzed at a suitable temperature. The toxicity risk of heavy metals confined in sludge SSB pyrolyzed around 600 °C could be decreased to very low.

## Figures and Tables

**Figure 1 materials-14-00447-f001:**
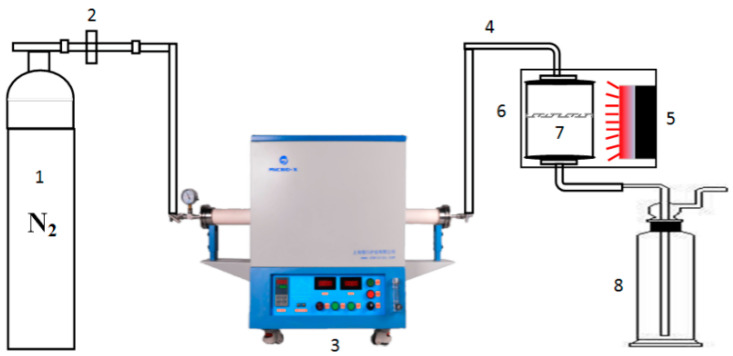
Diagram of self-made pyrolysis device. 1: Nitrogen, 2: Rotor flow, 3: Tube furnace, 4: Heating tube, 5: Sand bath, 6: Particle sampler, 7: Quartz filter, 8: Wash gas bottles.

**Figure 2 materials-14-00447-f002:**
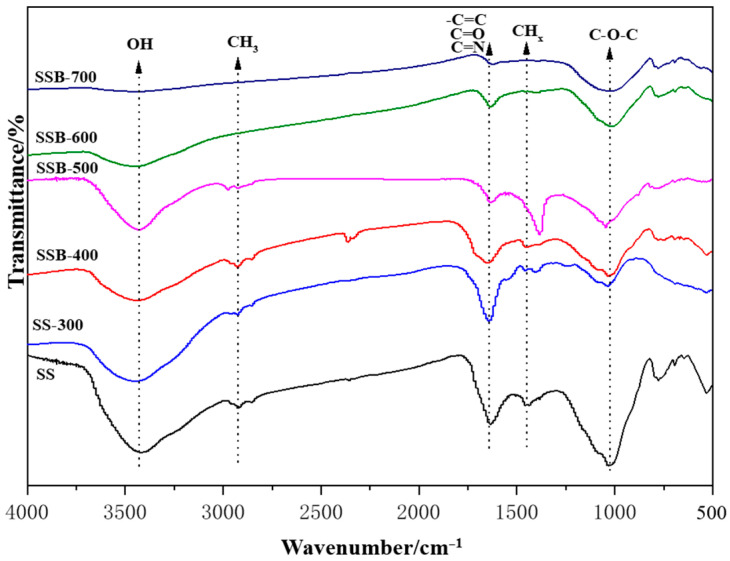
FTIR spectrum of the SS and its biochars. SS, sewage sludge; SSB-X, biochar prepared by pyrolysis of sewage sludge at X temperature (°C).

**Figure 3 materials-14-00447-f003:**
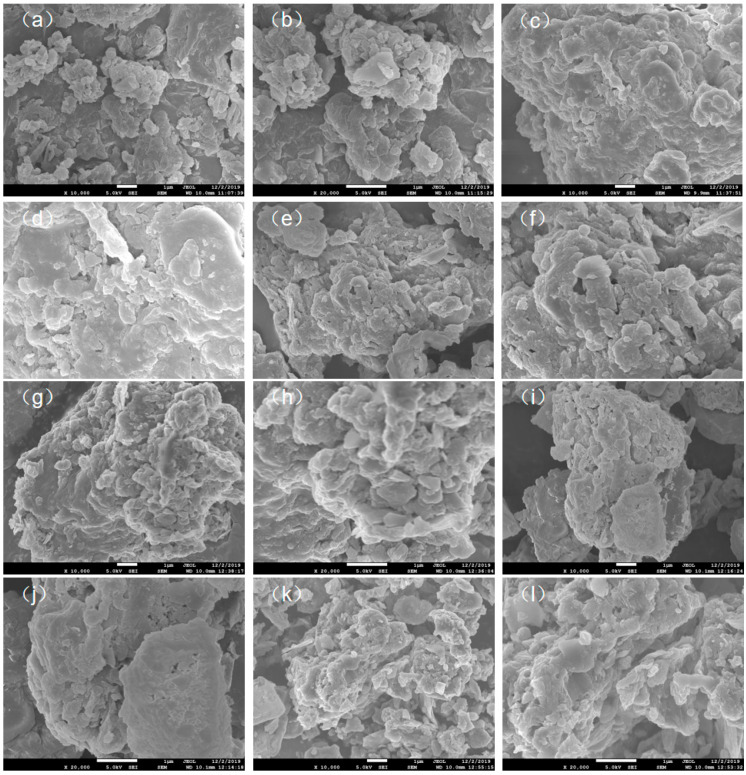
SEM images of sewage sludge and its biochar SSB. (**a**) SS(10,000×), (**b**) SS(20,000×), (**c**) SSB-300(10,000×), (**d**) SSB-300(20,000×), (**e**) SSB-400(10,000×), (**f**) SSB-400(20,000×) (**g**) SSB-500(10,000×), (**h**) SSB-500(20,000×), (**i**) SSB-600(10,000×), (**j**): SSB-600(20,000×), (**k**) SSB-700(10,000×), (**l**) SSB-700(20,000×).

**Figure 4 materials-14-00447-f004:**
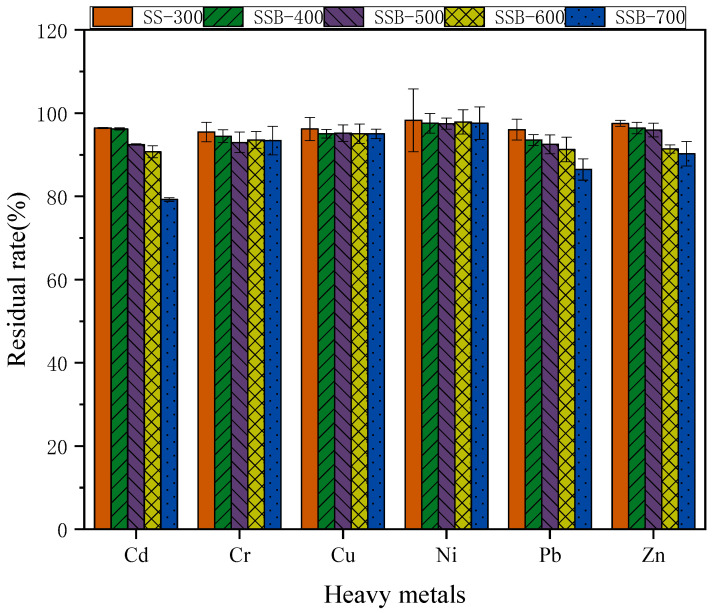
Residual rate of heavy metals in biochar at different pyrolysis temperatures.

**Figure 5 materials-14-00447-f005:**
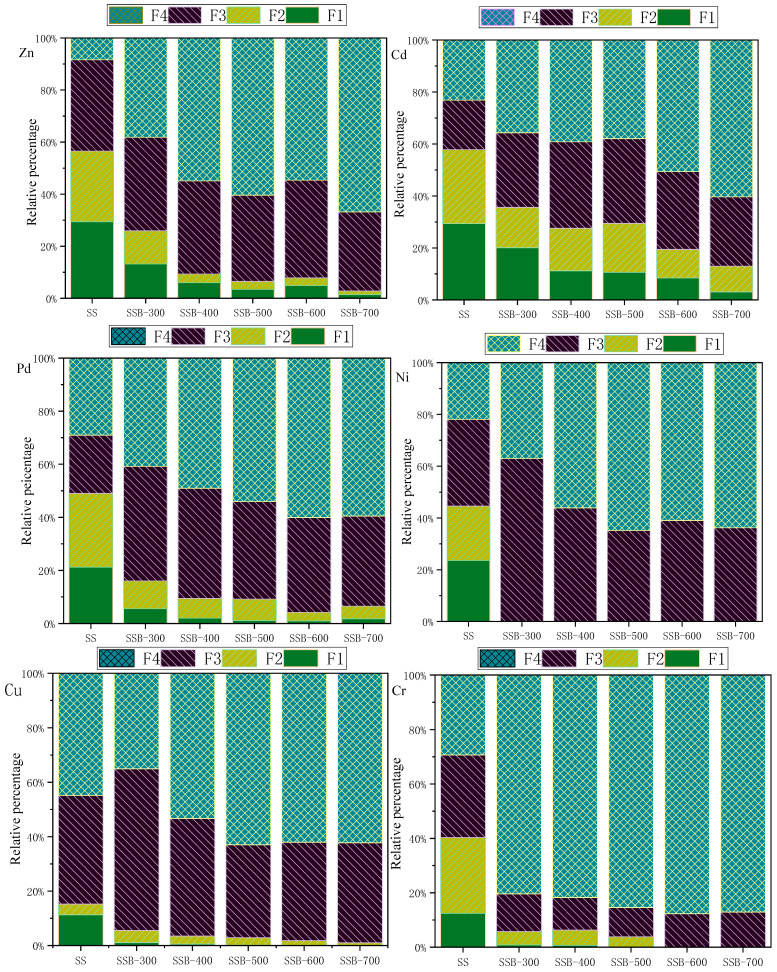
Chemical forms of heavy metals in SS and biochar.

**Figure 6 materials-14-00447-f006:**
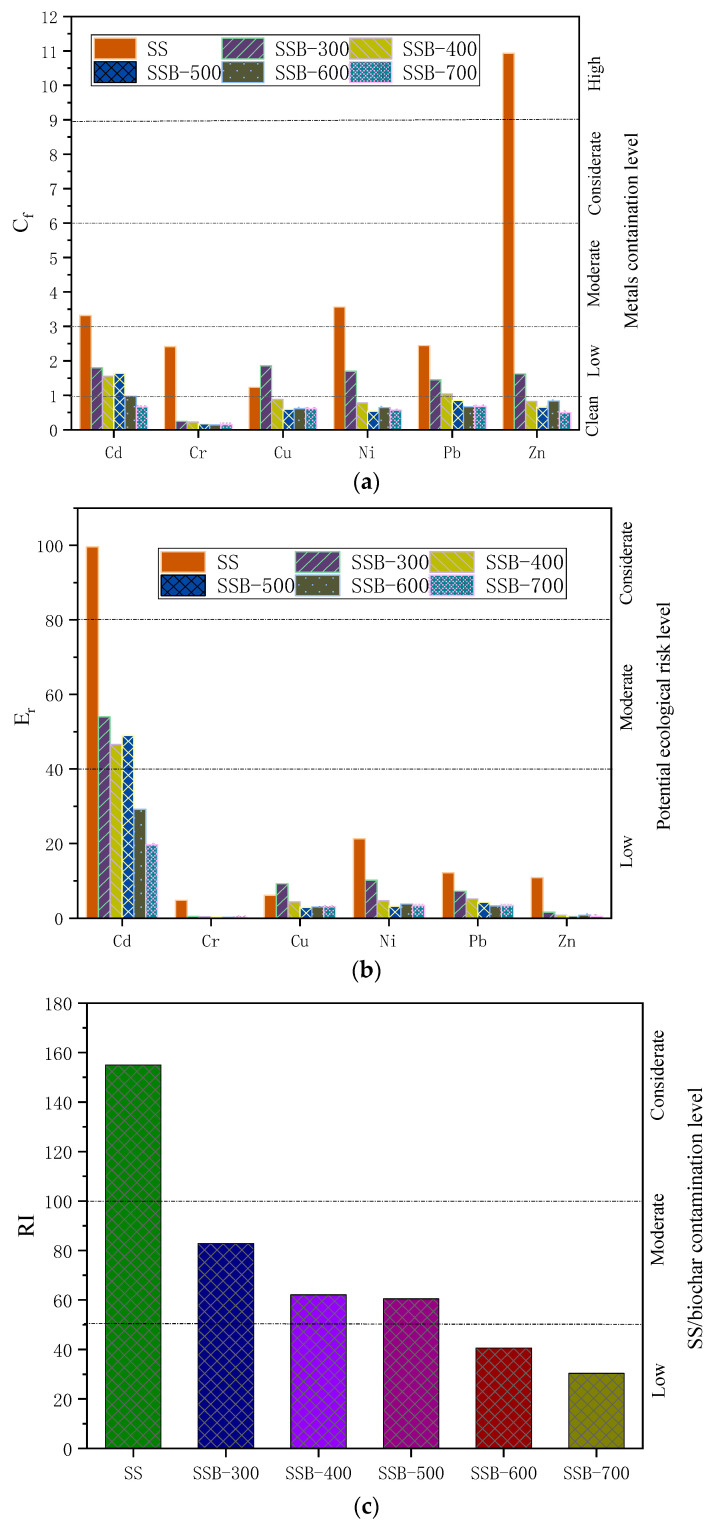
C_f_, E_r_, RI of SS and biochar. (**a**) Decrease of metals contamination/pollution factor with increase of pyrolysis temperatures (**b**) Decrease of metals ecological risk as pyrolyzed at rising temperature (**c**) Overall ecological risk index of various pyrolyzed biochar SSB.

**Table 1 materials-14-00447-t001:** Characteristics of the sewage sludge and sewage sludge biochar (SSB).

Characteristics	SS	SSB-300	SSB-400	SSB-500	SSB-600	SSB-700
Yield (wt %)	/	74.30 ± 0.43	63.58 ± 0.29	57.58 ± 0.29	55.42 ± 0.42	51.41 ± 0.08
Ash content (wt %)	38.85 ± 0.10	52.00 ± 0.38	63.62 ± 0.20	69.09 ± 0.40	71.20 ± 1.17	82.51 ± 0.45
pH value	6.77 ± 0.06	7.60 ± 0.00	7.73 ± 0.06	9.27 ± 0.06	10.27 ± 0.06	10.97 ± 0.06
C (wt %)	31.77 ± 0.04	31.55 ± 0.15	25.88 ± 0.15	22.66 ± 0.08	22.69 ± 0.08	13.69 ± 0.03
H (wt %)	4.30 ± 0.36	3.18 ± 0.17	1.83 ± 0.09	1.25 ± 0.10	0.58 ± 0.01	0.48 ± 0.00
N (wt %)	2.78 ± 0.06	2.68 ± 0.02	2.14 ± 0.03	1.83 ± 0.01	1.65 ± 0.04	0.76 ± 0.01
S (wt %)	0.61 ± 0.01	0.46 ± 0.01	0.27 ± 0.01	0.13 ± 0.00	0.11 ± 0.01	0.53 ± 0.01
O (wt %)	21.69	10.13	6.26	5.04	3.77	2.03
Molar H/C	1.62	1.21	0.85	0.66	0.31	0.42
Molar O/C	0.51	0.24	0.18	0.17	0.12	0.11
Specific surface area (m^2^/g)	1.88	2.03	3.42	5.88	11.16	11.26

SS, sewage sludge; SSB-X, SSB prepared by pyrolysis of sewage sludge at X °C.

**Table 2 materials-14-00447-t002:** Total contents of heavy metals in sewage sludge and biochar SSB.

Sample	Heavy Metal (mg·kg^−1^)					
	Cd	Cr	Cu	Ni	Pb	Zn
SS	7.05 ± 0.01	124.15 ± 0.86	134.70 ± 0.88	14.29 ± 0.06	121.78 ± 0.23	1341.73 ± 6.25
SSB-300	9.15 ± 0.01	159.51 ± 2.88	174.40 ± 3.71	18.90 ± 1.08	157.42 ± 3.01	1761.42 ± 9.26
SSB-400	10.67 ± 0.02	184.45 ± 1.93	201.38 ± 1.41	21.93 ± 0.34	179.20 ± 1.59	2034.77 ± 18.45
SSB-500	11.32 ± 0.01	200.45 ± 3.07	222.69 ± 2.66	24.19 ± 0.19	195.63 ± 2.75	2235.49 ± 21.93
SSB-600	11.54 ± 0.10	209.58 ± 2.49	231.02 ± 3.17	25.24 ± 0.42	200.53 ± 3.59	2212.49 ± 13.13
SSB-700	10.87 ± 0.03	225.61 ± 4.26	248.96 ± 1.54	27.12 ± 0.65	204.86 ± 3.10	2355.45 ± 39.24

## Data Availability

The data presented in this study are available on request from the corresponding author.

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
