# Peer review of "Experimental Investigation into the Effect of Pyrolysis on Chemical Forms of Heavy Metals in Sewage Sludge Biochar (SSB), with Brief Ecological Risk Assessment"

_materials, 2021, doi:10.3390/ma14020447_

Round 1

Reviewer 1 Report

The authors in this work deal with a very important issue, namely the treatment of sewage sludge. They analyze the impact that the pyrolysis has on the environmental hazard of the treated residue. The conclusions, which are condensed in the Ecological Risk Assessment, are very interesting, because they highlight that the pyrolysis treatment is more effective in rendering the residues inert, when the treatment temperature is higher.

But in this other work [Jin J , Li Y , Zhang J , et al. Influence of pyrolysis temperature on properties and environmental safety of heavy metals in biochars derived from municipal sewage sludge. Journal of Hazardous Materials, 2016,320:417-126.] (however quoted), the same results are described. And even if the treated samples come from a different place, every aspect in this proposed work, is quite similar and does not bring any significant novelty.

it is certainly a good technical report, but it cannot become an article for a scientific journal without an original contribution.

Author Response

Dear reviewer,

Thanks very much for your patient reading and kindly suggestions.  

The reference you mentioned is a very good reference. It systematically studies the characteristics and ecological risk assessment of pyrolytic biochar when the pyrolysis temperature is 400℃-600℃. It is of great help to my article writing and experimental design. However, the core purpose of this experiment is to provide a theoretical basis for the cement kiln cooperative sludge disposal technology. It uses the residual heat of flue gas in cement plants to dry and pyrolyze the sludge. The sludge thus treated is then used to prepare cement clinker. The flue gas temperature at the end of the cement kiln is about 300℃, and the residual heat of the flue gas at the kiln head is about 600-700℃. If there is heavy metal pollution in sludge obtained by flue gas pyrolysis is the first problem to be solved, so the temperature of pyrolysis is 300-700℃. Then, according to the preliminary test results of the sludge, the types of heavy metals in accordance with this experiment were determined, which increased Cd, reduced the Mn. Moreover, dust will be attached to the flue gas. We need to know the content of heavy metals in the pyrolysis products after the introduction of pyrolysis sludge, which increases the determination of heavy metal residue rate. Lastly, the RI results show that 300℃ to 400℃ and 600 to 700℃, the data have changed significantly, which is very necessary to complete the change, which is different from the change trend of 400-600℃. In summary, this work can provide a theoretical basis for the problem of heavy metals in the process of sludge disposal.

If there is anything else which I should do for the paper or something else, please contact me freely.

Best regards!

Yours,

Binbin

Reviewer 2 Report

This study evaluated the physicochemical properties, and potential environmental risks of heavy metal content in the biochar derived from sewage sludge (SS) at different pyrolysis temperatures (300 - 700 oC). Material properties was systematically characterized, and the results was clearly presented. However, this is a routine study, the results can be anticipated and lack of novelty. Therefore, the significance and innovation points should be more emphasized. Some of my comments are as follows: - The introduction should clearly show the knowledge gaps identified based on a comprehensive literature review on the present topic. - Lines 50-52, "... sludge pyrolysis can completely kill...causing environmental pollution". Please provide relevant data or references to support this statement. - SEM images are not distinct enough. - How did the author verify the "non-porous structure" and "porous structure" of SS and biochar by SEM observation? -The authors suggested that "the use of biochar has a very low environmental risk", have the authors considered the generation of harmful gas during the pyrolysis of SS? - Lines 302-303, "Other researchers have drawn similar conclusions about the pyrolysis of SS at different temperatures", the authors need to provided the relevant references to supports the comment. - In the conclusion, please strengthen the innovation and significance of the results. It is recommended to use quantitative reason in this section. - Please suggest some potential application of the biochar derived form SS.

Author Response

Dear reviewer,

Thanks very much for your patient reading and kindly suggestions.  

Those comments are very helpful for revising and improving our paper, as well as the important guiding significant to another research. We have studied the comments carefully and made corrections which we hope meet with approval. The main corrections made in the manuscript and the responds to the reviewers’ comments are as follows.

Question1: The introduction should clearly show the knowledge gaps identified based on a comprehensive literature review on the present topic.

Response: Thank you very much for your suggestion, which has been modified in the introduction, and marked in red.

Question2: - Lines 50-52, "... sludge pyrolysis can completely kill...causing environmental pollution". Please provide relevant data or references to support this statement.

Response: Thank you for your very detailed suggestion, the expression has been modified and marked red.

Question3: SEM images are not distinct enough. - How did the author verify the "non-porous structure" and "porous structure" of SS and biochar by SEM observation?

Response: This is my mistake, in the arrangement of the picture set too small and has been enlarged and rearranged. After magnifying the picture, the pore structure can be seen clearly, and combined with the previous specific surface area data, the of pyrolysis at different temperatures on biochar structure is analyzed.

Question4: The authors suggested that "the use of biochar has a very low environmental risk", have the authors considered the generation of harmful gas during the pyrolysis of SS?

Response: This statement is not accurate enough to cause you a misunderstanding. It has been modified in the paper. The gas pollution in the preparation process is very complex, which is not within the scope of this experiment.

Question5: "Other researchers have drawn similar conclusions about the pyrolysis of SS at different temperatures", the authors need to provided the relevant references to supports the comment.

Response: Thank you very much for your detailed advice. Reference files have been added and marked red.

Question6: In the conclusion, please strengthen the innovation and significance of the results. It is recommended to use quantitative reason in this section. - Please suggest some potential application of the biochar derived form SS.

Response: Your comments are correct. The core purpose of this paper is to provide a theoretical basis for cement kiln cooperative sludge disposal technology. It uses the residual heat of flue gas in cement plants to dry and pyrolyze the sludge. The sludge thus treated is then used to prepare cement clinker. The flue gas temperature at the end of the cement kiln is about 300℃, and the residual heat of the flue gas at the kiln head is about 600-700℃. If there is heavy metal pollution in biochar obtained by flue gas pyrolysis is the first problem to be solved, so the temperature of pyrolysis is 300-700℃. There is a lack of research on the potential environmental risks of sludge pyrolysis products at 300-700℃. Therefore, it is necessary to study the existing chemical forms of heavy metals in biochar, to carry out ecological environment risk assessments to minimize the risk of heavy metals being released into the environment. This work can provide a theoretical basis for the problem of heavy metals in the process of sludge disposal. According to the above statement, the introduction and conclusion are revised and marked red.

Reviewer 3 Report

I want to congratulate the authors for their work.

General comments

The paper fits with the topics of the Materials Journal. The manuscript aim is declaring to be: study on the effect of pyrolysis temperature on the physicochemical, structural, and morphological properties of sludge biochar and evaluation of the total concentration, chemical forms, and potential.

The layout of the article is correct. The introduction contains the necessary information. The obtained results were discussed and chosen publications in this field were referred.

My overall opinion is good for this paper. However, I don't recommend to editor to accept the manuscript without minor revision. My observations are presented below:

  1. The paper is equilibrated as important sections.
  2. I don’t consider the aim of the paper a clear goal. I ask authors to express which is the aim of the evaluation realized during the manuscript.
  3. I ask authors to explain why you choose to introduce the SS sample in a plastic bag and place it into the dryer. Have they analyzed before if the plastic bag doesn’t have any structure modification during the dry process (temperature up to 105C), thus influencing the SS sample?
  4. Line 157 – Line 159 “51.41%. At the same time, the mass ratio of ash in SS increased from 38.85% to 82.51%. This is because 157 the organic matter in the sludge was gradually decomposed, which is consistent with other research 158 results.”

For this comment the authors may introduce some citations

  1. I ask authors to be more explicit in using the term “activated carbon” from phrase “However, the specific surface area of biochar was substantially smaller than that of activated carbon.” (lines: 177-178). This is the only time you mention these words.

Author Response

Dear reviewer,

Thanks very much for your patient reading and kindly suggestions.  

Those comments are very helpful for revising and improving our paper, as well as the important guiding significant to another research. We have studied the comments carefully and made corrections which we hope meet with approval. The main corrections made in the manuscript and the responds to the reviewers’ comments are as follows.

Question1: I don’t consider the aim of the paper a clear goal. I ask authors to express which is the aim of the evaluation realized during the manuscript.

Response: In the conclusion, please strengthen the innovation and significance of the results. It is recommended to use quantitative reason in this section. - Please suggest some potential application of the biochar derived form SS.

Response: Your comments are correct. The core purpose of this paper is to provide a theoretical basis for cement kiln cooperative sludge disposal technology. It uses the residual heat of flue gas in cement plants to dry and pyrolyze the sludge. The sludge thus treated is then used to prepare cement clinker. The flue gas temperature at the end of the cement kiln is about 300℃, and the residual heat of the flue gas at the kiln head is about 600-700℃. If there is heavy metal pollution in biochar obtained by flue gas pyrolysis is the first problem to be solved, so the temperature of pyrolysis is 300-700℃. There is a lack of research on the potential environmental risks of sludge pyrolysis products at 300-700℃. Therefore, it is necessary to study the existing chemical forms of heavy metals in biochar, to carry out ecological environment risk assessments to minimize the risk of heavy metals being released into the environment. This work can provide a theoretical basis for the problem of heavy metals in the process of sludge disposal. According to the above statement, the introduction and conclusion are revised and marked red.

Question2: I ask authors to explain why you choose to introduce the SS sample in a plastic bag and place it into the dryer. Have they analyzed before if the plastic bag doesn’t have any structure modification during the dry process (temperature up to 105℃), thus influencing the SS sample?

Response: The pretreatment process of sludge mainly refers to the pretreatment of coal pyrolysis. Design is mainly to study the process of sludge pyrolysis, need to eliminate the effect of moisture drying, so select drying temperature of 105℃.

Question3: Line 157 – Line 159 “51.41%. At the same time, the mass ratio of ash in SS increased from 38.85% to 82.51%. This is because 157 the organic matter in the sludge was gradually decomposed, which is consistent with other research 158 results.”

Response: Thank you for your detailed advice. The document has been added and marked with red handwriting

Question4: I ask authors to be more explicit in using the term “activated carbon” from phrase “However, the specific surface area of biochar was substantially smaller than that of activated carbon.” (lines: 177-178). This is the only time you mention these words.

Response: This statement is not clear enough, and has nothing to do with the main purpose of the article, has been deleted in the article, thank you again for your careful review

If there is anything else which I should do for the paper or something else, please contact me freely.

Best regards!

Yours,

Binbin

Round 2

Reviewer 1 Report

With the changes made by the authors, the work reaches a sufficient level of originality, therefore I recommend publication in the present form.

Reviewer 2 Report

The authors have made great efforts to improve this manuscript. I have no further comment. The revised manuscript can be considered for publication.